# Interfacial Stresses of Thermal Barrier Coating with Film Cooling Holes Induced by CMAS Infiltration

**Chenchun Chiu [1], Shaochen Tseng [1], Chingkong Chao [1,*], Xueling Fan [2,*] and Weihung Cheng [1]**

1 Department of Mechanical Engineering, National Taiwan University of Science and Technology, Taipei 106335, Taiwan; d11003007@mail.ntust.edu.tw (C.C.); d10603001@mail.ntust.edu.tw (S.T.); wayhome723@gmail.com (W.C.)
2 Joint Research Center for Extreme Environment and Protection Technology, School of Aerospace Engineering, Xi'an Jiaotong University, Xi'an 710049, China
* Correspondence: ckchao@mail.ntust.edu.tw (C.C.); fanxueling@mail.xjtu.edu.cn (X.F.)

**Abstract:** To obtain high gas turbine efficiency, a film cooling hole is introduced to prevent the destruction of thermal barrier coating systems (TBCs) due to hot gases. Furthermore, environmental calcium-magnesium-aluminum-silicate (CMAS) particulates plug the film cooling hole and infiltrate the TBCs to form a CMAS-rich layer, which results in phase transformations and significant modifications in the thermomechanical properties that impact the TBCs during cooling. This study aimed to establish a three-dimensional thermo-fluid-solid coupling TBCs model with film cooling holes and CMAS infiltration to analyze the temperature and residual stress distribution via simulations. For the interfacial stress around the cooling hole at the TC/BC interface, the film cooling holes alleviated the interfacial residual stress by 60% due to the reduction in temperature by 40%. In addition, CMAS infiltration intensified the interfacial residual stress via phase transformation. As a result of the influence of larger penetration depths and expansion rates of phase transformation, a significant increase in residual stress was observed. At the beginning of CMAS infiltration, the interfacial stress would be more dominated by the effect of infiltration depth. In addition, the failure due to interfacial normal and tangential stresses was more likely to be found at the infiltration zone near the cooling hole.

**Keywords:** thermal barrier coating; film cooling hole; phase transformation; calcium-magnesium-aluminum-silicate (CMAS); thermal-fluid-solid coupling

## 1. Introduction

Innovations in gas turbines have been widely implemented in aviation systems over the past few decades. Turbine blade components generally operate at elevated temperatures and are subjected to high thermal loads. Hence, in recent research, thermal barrier coating systems (TBCs) have been considered to provide thermal protection. TBCs are multilayer systems that include a top coat (TC) layer, a bond coat (BC) layer, a thermally grown oxide layer, and a substrate. It helps protect the substrate from hot corrosion and thermal loads owing to high temperature gradients. In addition, it enables the gas turbine to operate at gas temperatures higher than those without TBCs [1,2]. For better efficiency, a film cooling hole is introduced in TBCs [3]. A coolant is introduced through the film cooling hole into the surface of the airfoil to decrease the temperature [4]. Therefore, in addition to the increase in turbine efficiency, the presence of film cooling holes protects the TBCs from destruction caused by hot gases.

The operation of a gas power turbine is affected when environmental particulates such as volcanic ash, sand, and fly ash accumulate on the surface [5,6]. These particulates generally have complex chemical composition and a wide range of melting temperatures. The complicated constituents generally comprise calcium-magnesium-aluminum-silicates (CMAS). In addition to the complexity of the constituents, CMAS may plug the film cooling

hole [7] and attack the TBCs by infiltrating the pores, which would cause changes in the thermomechanical properties of the TC layer [8]. Owing to the difficulty in experimentally measuring the exact magnitudes of the material properties, only the effective material characteristics of the infiltration layer were determined by the theoretical method [9].

In addition, CMAS infiltration played a significant role in the phase transformation of yttria-stabilized zirconia (YSZ). Ideally, 7%–8% $Y_2O_3$ is used in $ZrO_2$ (TC layer) to form a metastable phase (t'). The t' phase does not undergo phase transformations when the temperature varies [2]. However, Fang et al. [10] and Clarke et al. [1] showed that CMAS starts to melt at 1089–1224 °C and infiltrates into the TC layer because of capillary action to form two different layers. One is a CMAS-rich layer, and the other is a non-CMAS layer (original 7YSZ). At elevated temperatures, the dissolution of yttrium in the CMAS-rich layer occurs because of the chemical reactions between CMAS and YSZ. Therefore, the CMAS-rich layer becomes an yttrium-depleted layer owing to the decrease in the volume fraction of yttrium. In addition, the CMAS-rich layer causes a 3%–5% volume expansion owing to martensitic transformation (tetragonal to monoclinic) during the cooling period [11], which results in an increase in interfacial stress and delamination of the TBCs as a consequence of volume expansion. Hence, CMAS infiltration affects the stability of TBCs.

Owing to the complicated constituents of the TBCs, several factors may be responsible for causing premature fracture, including large residual stress due to thermal and mechanical mismatch [12] and thermal stress due to the large thermal temperature gradient when the engine shuts down [13]. Furthermore, the CMAS-rich layer within the TC layer is a dominant factor because the interfacial stress is intensified owing to phase transformation [14]. The film cooling hole causes thermal load in the TBCs owing to the large temperature gradients [15]. In addition, stress concentration is observed near the film cooling hole [16,17]. Jiang et al. [18] and Meng et al. [19] indicated that the free edge effect of the film cooling hole is important for the analysis of interfacial stress. The interfacial peeling stress and shear stress between the layers lead to mode-I and mode-II delamination, respectively. In addition, in the analytical solution, a stiffer topcoat increases the peeling stress and shear stress at all the interfaces. Such elastic mismatches between the different layers promote edge delamination at the interface [20,21]. Hence, the stiffer TC layer due to the CMAS infiltration and phase transformation is expected to impact the interfacial stress that is required for the failure analysis of a TBCs with film cooling holes. In recent years, most studies mainly focused on the cooling performance with different shapes and angles of the cooling hole using the simulation method [22–25]. However, few studies have investigated the thermo-fluid-solid interaction between these two factors (film cooling hole and CMAS) [26]. Dai et al. [26] primarily investigated the temperature field under the different shape of cooling hole for preventing CMAS infiltration; nevertheless, the fracture behavior of TBCs via interfacial stresses owing to the CMAS infiltration was not considered. Thus, the present study proposed thermo-fluid-solid coupling analysis of TBCs with film cooling holes, and volcano-CMAS infiltrations have been discussed using a simulation method.

This study investigated the evolution of interfacial residual stress near a round cooling hole with a 30° inclined angle caused by CMAS infiltration during the cooling period. This research was primarily discussed using the finite element method. The thermo-fluid-solid couple analysis was extremely difficult compared to structural analysis. Except for the interaction between the flow and thermal field, the film cooling hole resulted in a complex temperature distribution with large temperature gradients. In addition, based on the temperature distribution, it was necessary to control CMAS infiltration using the ABAQUS subroutine at a specific temperature. Hence, the analysis of interfacial residual stress became more challenging. First, the thermal boundary conditions were solved using ANSYS FLUENT. Moreover, the CMAS infiltration region was defined based on the previous temperature distribution. Finally, the stress field in the solid domain was numerically calculated using ABAQUS.

## 2. Modeling Methodology

Thermo-fluid-solid couple analysis was performed using two commercial software packages (ANSYS 2020 R2 and ABAQUS 6.14). According to a previous study [16], an outline of the coupling analysis of the TBCs with film cooling holes and CMAS infiltration is shown in Figure 1a. In this study, the temperature distribution induced by the heat fluid was numerically calculated using ANSYS Fluent. Thereafter, this distribution was assumed as a boundary condition and imported into the finite element program. Finally, the residual stress distribution was obtained using ABAQUS at room temperature. In this simulation, three-dimensional fluid and solid models were investigated as follows.

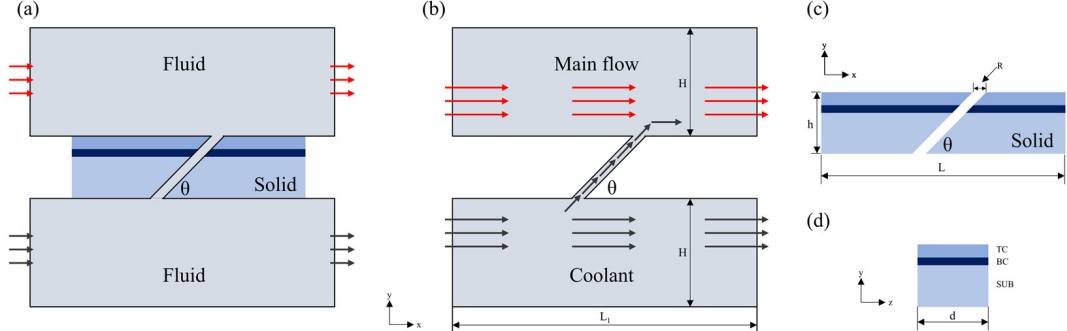

**Figure 1.** (**a**) Outlined photograph for TBCs with film cooling hole combined with fluid and solid domain, (**b**) fluid domain in xy plane, (**c**) solid domain in xy plane, and (**d**) yz plane.

### 2.1. CFD Modeling of the Fluid Domain

The three-dimensional fluid domain model is shown in Figure 1b. The size of both regions was 60 mm (L1) × 12 mm (H), and the width of the model was 10 mm (z direction). The boundary and geometry of the fluid domain were studied by Jiang et al. [16]. The upper and lower regions were the mainstream and coolant fluids, respectively. For both the mainstream and coolant fluids, the fluid velocity and pressure were assumed to be 128 m/s and 15 atm, respectively [27]. The temperature of the fluid was 1400 or 500 °C for the mainstream and coolant fluids, respectively [27]. The inlet and outlet surfaces set up the periodic boundary conditions, and the two side surfaces were assigned symmetric conditions. The remaining surface was assigned a no-slip velocity boundary condition and was adiabatic. The fluid domain consisted of 500,000 elements. In ANSYS Fluent, the simulation could numerically calculate the velocity and temperature field for a wide variety of fluids with different assumptions such as incompressible and compressible, laminar and turbulent under the steady and transient analysis [27–30]. In the present study, the flow was assumed to be compressible and turbulent under the steady state condition. Hence, the basic governing equations were based on the continuity equation as follows [29]:

$$\frac{\partial}{\partial X_i}(\rho u_i) = 0 \tag{1}$$

where $u_i$ represents the mean velocity, and $\rho$ is the density. In addition, $X_i$ stands for the coordinates x, y, and z when $i$ is equal to 1, 2, and 3, respectively. In addition, Reynolds-averaged Navier–Stokes (RANS) equations would be constructed for the calculation of turbulent as follows [29]:

$$\frac{\partial}{\partial X_j}(\rho u_i u_j) = -\frac{\partial \rho}{\partial X_i} + \frac{\partial}{\partial X_j}[\mu(\frac{\partial u_i}{\partial X_j} + \frac{\partial u_j}{\partial X_i} - \frac{2}{3}\delta_{ij}\frac{\partial u_k}{\partial X_k})] + \frac{\partial}{\partial X_j}(-\rho \overline{u'_i}\overline{u'_j}) \tag{2}$$

where $u'$ indicates the fluctuating velocity. For the heat transfer and compressibility of flow, energy equations were introduced as follows [29]:

$$\frac{\partial}{\partial X_j}[u_i(\rho E + p)] = \frac{\partial}{\partial X_j}[(k + \frac{C_p \mu_t}{0.85})\frac{\partial T}{\partial X_j} + u_i(-\rho \overline{u}_i' \overline{u}_j')] \tag{3}$$

In addition, $p$ and $\mu$ are pressure and viscosity, respectively. The final term $\rho \overline{u}_i' \overline{u}_j'$ in Equation (2) is the Reynolds stress component. To find the stress component, Boussinesq approximation was introduced as following [29]:

$$-\rho \overline{u}_i' \overline{u}_j' = \mu_t(\frac{\partial u_i}{\partial u_j} + \frac{\partial u_j}{\partial u_i}) - \frac{2}{3}(\rho k + \mu_t \frac{\partial u_k}{\partial u_k})\delta_{ij} \tag{4}$$

where $k$ and $\mu_t$ represent the turbulent kinetic energy and turbulent viscosity, respectively. For calculating these two terms ($k$ and $\mu_t$), SST-$k\omega$ equations were proposed as follows [29]:

$k$-equation

$$\frac{\partial}{\partial X_j}(\rho u_j k) = \frac{\partial}{\partial X_j}[(\mu + \frac{\mu_t}{\sigma_k})\frac{\partial k}{\partial X_j}] + \rho(P_k^* - \beta_1 \varepsilon - \beta_2 k\omega) \tag{5}$$

$\omega$-equation

$$\frac{\partial}{\partial X_j}(\rho u_j \omega) = \frac{\partial}{\partial X_j}[(\mu + \frac{\mu_t}{\sigma_\omega})\frac{\partial \omega}{\partial X_j}] + \rho^2 \frac{\gamma_1}{\mu_t} P_k - \rho \beta_3 \omega^2 + F_{SST} \tag{6}$$

where $\varepsilon$ is the turbulent energy dissipation; additionally, $\omega$ represents the rate at which turbulence kinetic energy is converted into thermal internal energy per unit volume and time. According to the previous research [31,32], $\beta_1$, $\beta_2$ and $\beta_3$, are 0.075, 0.09, and 0.0828, respectively. Besides, $\gamma_2$ and $\kappa$ are 0.44 and 0.41; $\sigma_k$ and $\sigma_\omega$ are 1 and 0.857.

*2.2. FE Modeling of the Solid Domain*

In this study, the solid model comprised a TC layer, a BC layer, and substrate, as shown in Figure 1c, in which the effect of cooling period was primarily discussed and the influence of TGO was ignored. The thicknesses of the TC, BC, and substrate were 0.35, 0.15, and 3.5 mm, respectively. Hence, the total thickness of the solid domain was 4 mm. A round film cooling hole with a diameter (R) of 1 mm was embedded in the solid domain at an inclined angle of 30°. The width (d) of the solid domain was 3 mm, and the length (L) parallel to the mainstream direction was 40 mm, as shown in Figure 1c,d. According to the temperature field calculated by ANSYS Fluent, the model was divided into two zones for simulating CMAS infiltration, as shown in Figure 2a. One was infiltration zone, and the other was non-infiltration zone. The element in the infiltration zone under high temperature was assumed to undergo CMAS infiltration. The realistic model was simplified into periodic and symmetric models. Both the right and left surfaces in the x-direction had established periodic boundary conditions, and the bottom surface was fixed. The remaining surface and the cooling hole set up the boundary condition of traction-free. To understand the role of phase transformation, this study solely considered the cooling process for performing the simulation. To determine the boundary conditions for temperature, the solid model was simulated from the initial operating condition that was imported by the fluid domain and then linearly cooled down to room temperature for 10 min. The present study set up at a stress-free state such that the creep phenomenon at the initial stage would relieve the stress. As a result, stress was lower during the operating period compared to the residual stress during the cooling period. That is, this study focused on the residual stress to analyze the fracture behavior of the TBCs. The solid domain comprised 200,000 elements, and the mesh types were DC3D8 and DC3D6, as shown in Figure 2b. The mesh near the cooling hole was significantly refined to ensure the accuracy of the results, as shown in Figure 2c,d. The error of convergence analysis for the stress field was less than 1% in this study.

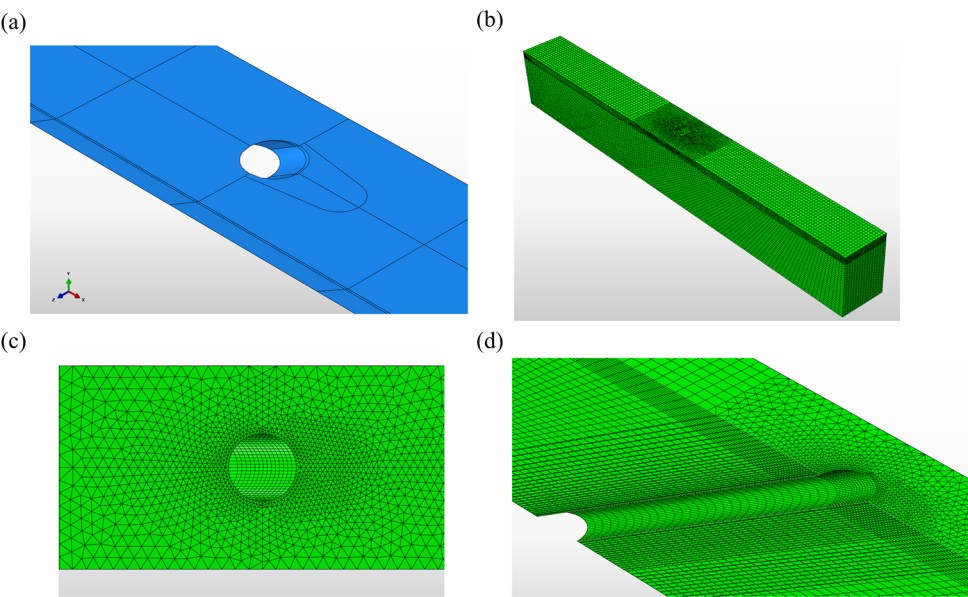

**Figure 2.** Outlined photograph for (**a**) dividing the solid model into infiltration and non-infiltration zones and meshing (**b**) whole solid domain, (**c**) solid domain in xz plane, and (**d**) half the solid domain.

### 2.3. Material Characteristic

In the present study, the thermomechanical material properties of the multilayers of the TBCs are listed in Table 1 [17,33–35]. As shown, these properties were temperature-dependent, which indicated that the material properties displayed a linear behavior owing to a decrease in temperature.

**Table 1.** Thermomechanical material properties for each layer of TBCs [17,33–35].

| Material | T (°C) | E (GPa) | v | k (W/mK) | $\alpha \times 10^{-6}$ (°C$^{-1}$) |
|---|---|---|---|---|---|
| TC layer | 25 | 17.5 | 0.2 | 1.05 | 9.68 |
|  | 1000 | 12.4 |  | 1.88 | 11 |
| Bond coat | 25 | 220 |  | 4.3 | 10.3 |
|  | 400 | 200 | 0.3 | 6.4 | 12.7 |
|  | 800 | 164 |  | 10.2 | 14.1 |
|  | 1000 | 120 |  | 11.1 | 20.4 |
| Substrate | 100 | 209 | 0.38 | 11.4 | 11.1 |
|  | 300 | 199 | 0.38 | 14.9 | 13.3 |
|  | 500 | 185 | 0.39 | 18.3 | 14.0 |
|  | 700 | 167 | 0.39 | 21.8 | 14.6 |
|  | 900 | 145 | 0.4 | 25.2 | 15.4 |

In addition to the influence of temperature, the presence of CMAS infiltration also played a vital role in the material properties. Specifically, the effect of CMAS infiltration was enhanced when the penetration depth within the TC layer increased. From a previous study [36], it was shown that molten CMAS penetrated into the TC layer to form an infiltration layer. Hence, the magnitude of Young's modulus, coefficient of thermal expansion (CTE), and thermal conductivity with regard to this infiltration layer were modified to their effective values by formulation because it was difficult to measure them experimentally. The following equation was the effective Young's modulus as a function of the volume fraction of the infiltration layer [9]:

$$E_{Eff} = \left(\frac{V_{dense-YSZ}}{E_{dence-YSZ}} + \frac{V_{CMAS}}{E_{CMAS}}\right)^{-1} \tag{7}$$

where $E_{dense-YSZ}$ and $E_{CMAS}$ denote the elastic moduli of dense t'-YSZ and CMAS, respectively. $V_{YSZ}$ (90%) and $V_{CMAS}$ (10%) were the volume fractions of the TC layer and CMAS, respectively. Levi et al. [37] indicated that Young's modulus of YSZ drastically increases owing to the stiffening caused by CMAS penetration, which implies that the material characteristics of YSZ would transfer from a softer material to a harder material owing to CMAS penetration. Hence, Young's modulus of YSZ stiffening was defined as 200 GPa, as summarized in Table 2 [38,39]. Based on Equation (1), the effective Young's modulus for the infiltration layer was calculated to be 175 GPa, as listed in Table 2.

**Table 2.** Material properties of dense YSZ, CMAS, and infiltration layer [38,39].

| Material | E (GPa) | v | k (W/mK) | $\alpha \times 10^{-6}$ (°C$^{-1}$) |
|---|---|---|---|---|
| Dense YSZ | 200 | 0.2 | 2.5 | 11 |
| CMAS | 80 | 0.25 | 1.78 | 8.1 |
| Infiltration layer (Dense YSZ + CMAS) | 175.73 | 0.2 | 2.33 | 10.87 |

The effective CTE of the infiltration layer was modified to an effective value using the following equation [9]:

$$\alpha_{Eff} = \frac{\alpha_{CMAS}E_{CMAS}V_{CMAS} + \alpha_{dense-YSZ}E_{dense-YSZ}V_{dense-YSZ}}{E_{CMAS}V_{CMAS} + E_{dense-YSZ}V_{dense-YSZ}} \tag{8}$$

In this study, the $\alpha_{dense-YSZ}$ and $\alpha_{CMAS}$ were defined as $11 \times 10^{-6}$ (°C$^{-1}$) and $8.1 \times 10^{-6}$ (°C$^{-1}$), as listed in Table 2; furthermore, the calculated effective CTE of the infiltration layer was $10.87 \times 10^{-6}$ (°C$^{-1}$). The effective coefficient of thermal conductivity of the infiltration layer was also transformed to an effective value using the following equation [9]:

$$k_{Eff} = k_{dense-YSZ}(1 + V_{CMAS}(\gamma - 1)) \tag{9}$$

where $\gamma = k_{CMAS}/k_{dense-YSZ}$, and the values of $k_{dense-YSZ}$ and $k_{CMAS}$ were defined as 1.05 (W/mK) and 1.78 (W/mK), respectively, as listed in Table 2. The magnitude of $k_{CMAS}$ was determined for crystallized CMAS [38]. The calculated effective coefficient of thermal conductivity of the infiltration layer was 2.33 (W/mK), as listed in Table 2.

*2.4. Volume Expansion by Phase Transformation*

From experimental observations [11], there were several factors responsible for causing volume changes in the TBCs, such as infiltration of CMAS, dissolution, sintering, phase transformation of 7YSZ, and crystallization of molten CMAS. Specifically, the infiltration of CMAS, sintering effect, and dissolution of 7YSZ occurred at elevated temperatures. However, the phase transformation of 7YSZ and crystallization of molten CMAS occurred during the cooling period. In addition, this study considered only the cooling process. Hence, only the phase transformation of 7YSZ was investigated. According to the experimental results [10], CMAS started to crystallize in the range of 900–1000 °C and melted in the range from 1089 to 1224 °C. In this temperature range (1089–1224 °C), the 7YSZ TBCs were gradually infiltrated by the molten CMAS. Hence, in this study, the infiltration temperature was assumed to be 1080 °C. Accordingly, in the simulation, when the temperature was higher than 1080 °C, it was defined as an infiltration site. After the infiltration of CMAS, the TC layer was divided into two different layers in the xy plane. One was a CMAS-rich layer, and the other was a non-CMAS layer. The CMAS-rich layer played a significant role in phase transformations. Garces et al. [40] indicated that Raman mapping indicates that the volume fraction of m-YSZ increases during the cooling period. This phenomenon

occurred owing to the chemical reactions between the CMAS and 7YSZ, which led to a decrease in the volume fraction of yttrium in 7YSZ. Hence, the shortage of yttrium caused the CMAS-rich layer to become an yttrium-depleted layer that resulted in the martensitic transformation from the tetragonal to the monoclinic phase during the cooling period. In addition, the most vital finding was that the martensitic phase transformation of YSZ caused a 3%–5% volume expansion in the TC layer. Hence, it was concluded that volume expansion occurred in the CMAS-rich layer owing to the phase transformation.

In addition, Yashima et al. [41] determined that the transition temperature depends on the concentration of $Y_2O_3$. Regarding the Raman spectrum of the 7YSZ TBC sample at each temperature [40], the characteristic peaks of the monoclinic phase are observed at approximately 600 °C. Therefore, in this study, the range of transition temperature was assumed to be 500–600 °C. To be more specific, the CMAS-rich layer began to undergo a phase transformation at 600 °C and exhibited a complete transition at 500 °C, which indicated that the infiltration layer was forcibly implemented for volume expansion via phase transformation at the transition temperature (from 600 to 500 °C). Moreover, the volume expansion by CMAS infiltration was simulated by a subroutine in ABAQUS (UEXPAN). In the simulation, the expansion in the element was assumed to be isotropic. Therefore, the element expanded equally along the width (d), lateral (L), and depth (h) directions.

## 3. Results

### 3.1. Influence of Film Cooling Holes

The presence of film cooling holes was used to increase the efficiency of the gas turbines. However, film cooling had a crucial influence on the temperature and stress fields in a TBCs. Initially, the temperature distributions of TBCs without and with film cooling holes are shown in Figure 3a,b, respectively. The temperature distribution at the initial stage was similar to that in the reference study [17], as shown in Figure 3c. The temperature distribution of a TBCs with a film cooling hole had a larger temperature gradient and more complex distribution compared to those of TBCs without a cooling hole. This indicates that the TBCs with film cooling holes had a large thermal mismatch as a result of the large temperature gradient. The interface between the BC and TC layers is the most vulnerable layer of the TBCs, resulting in premature fractures and delamination. Hence, in this study, the BC top surface was primarily considered. The temperature distribution at the initial stage and von Mises stress distribution during the cooling period along the BC top surface without and with film cooling holes are shown in Figures 4a and 3b, respectively. The solid and dashed lines represent the TBCs without and with film cooling holes, respectively. The investigation helped demonstrate that the presence of a film cooling hole affected the temperature distribution. The TBCs without a cooling hole presented a uniform temperature distribution, whereas the TBCs with a cooling hole showed that the magnitude of temperature decreased near the cooling hole, as shown in Figure 4a. Specifically, the temperature was more alleviated when it was closer to the cooling hole. In addition, this study set up the stress-free state at elevated temperatures, which meant that at the initial stage, the stress was zero. When the temperature cooled down to room temperature (25 °C), the residual stress played a dominant role in the delamination of TBCs [17,42,43]. Hence, in the following discussion, the present study only considered the residual stress during the cooling period. The von Mises stress distributions during cooling period along the BC top surface without and with the film cooling hole are shown in Figure 4b. As expected, the von Mises stress distribution was significantly influenced by the cooling hole. The TBCs with cooling holes demonstrated that the stress magnitude was greatly alleviated near the cooling hole, as shown in Figure 4b. As can be seen, the temperatures without and with the cooling hole were 900 and 500 °C, respectively, as shown in Figure 4a. Once the TBCs cooled down to room temperature (25 °C), the latter exhibited a smaller difference in temperature, which resulted in lower stress, as shown in Figure 4b. In addition, it was interesting to observe that stress concentration occurred near the film cooling hole. Jiang et al. [15,16] also indicated that residual stress drastically

decreases because of the film cooling hole. These results could be explained by the fact that the presence of a cooling hole suppresses the residual stress. It was concluded that the presence of a cooling hole would suppress the interfacial residual stress by 60% due to decreasing the temperature by 40%.

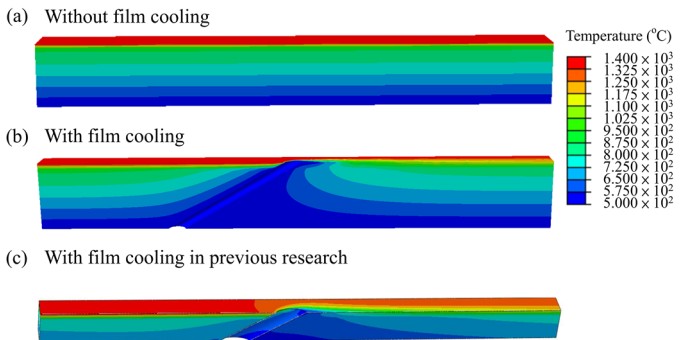

**Figure 3.** Temperature distribution (**a**) without film cooling, (**b**) with film cooling, and (**c**) compared to previous result [17].

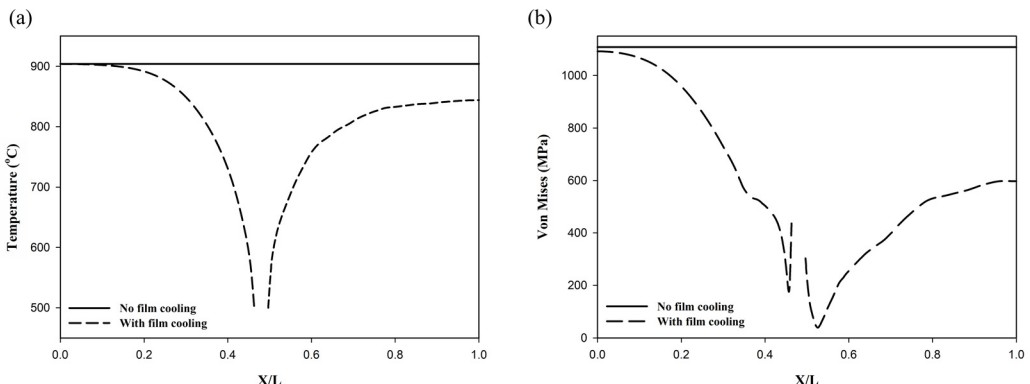

**Figure 4.** (**a**) Temperature distribution along the BC top surface at the initial stage and (**b**) von Mises stress distribution along the BC top surface during cooling time.

### 3.2. Influence of CMAS Infiltration

A schematic of the infiltration region on the TC top surface according to the temperature assumptions of CMAS infiltration at the initial stage is shown in Figure 5a. Specifically, the labels A and B around the cooling hole represent the locations near the non-infiltration zone and infiltration zone, respectively. The solid line represents the boundary between the non-infiltration and infiltration zones in the xz plane. To clarify the CMAS infiltration zone, the temperature distribution along the dashed line denoted in Figure 5a at initial time is shown in Figure 5b. As we can see, the trend of temperature on the TC top surface was similar to that on the BC top surface, as shown in Figure 4a. However, the temperature gradient near the cooling hole was different because heat conduction had not occurred on the TC surface as shown in Figure 5b. As previously defined, the infiltration temperature was assumed to be 1080 °C, which was attributed to the melting temperature. As shown in Figure 5b, the temperature in the infiltration zone and non-infiltration zone was defined to be higher and lower than the infiltration temperature, respectively. Based on these hypotheses, the temperature inside the boundary was lower than the infiltration temperature as a consequence of the film cooling hole, which implied that the region inside was not infiltrated by CMAS. However, the region outside the boundary was defined as an infiltration zone. Next, we confirmed the influence of the phase transformation at the infiltration zone. The overall temperature distribution on the TC top surface at 350 s and 400 s during the cooling period is shown in Figure 6a,b, respectively. As shown, the highest temperatures at 350 and 400 s were almost 600 and 500 °C during the cooling

period, respectively. To be more specific, the temperature distribution along the dashed line denoted in Figure 6a,b is shown in Figure 6c. The results showed that the temperature of the infiltration zone was higher than 600 °C at 350 s, which indicated the occurrence of phase transformation. Moreover, the temperature of the infiltration zone was lower than 500 °C at 400 s, which indicated the completion of the phase transformation. Hence, the influence of the phase transformation at the infiltration zone was verified to occur during this period (350–400 s).

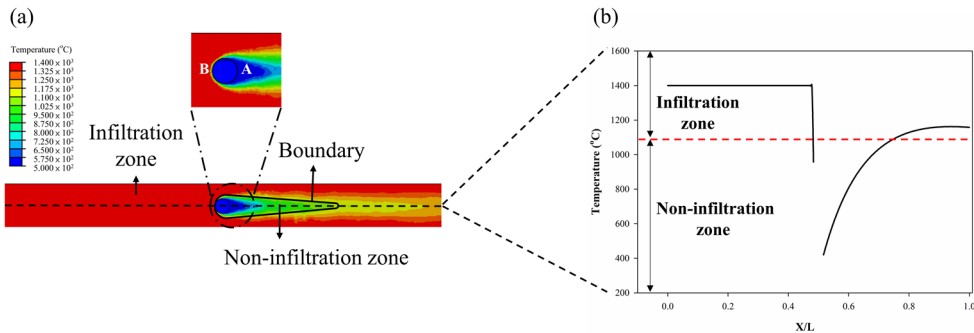

**Figure 5.** (**a**) Outlined photo for illustrating the infiltration zone and the overall temperature distribution and (**b**) the temperature distribution along the dashed line at the initial stage.

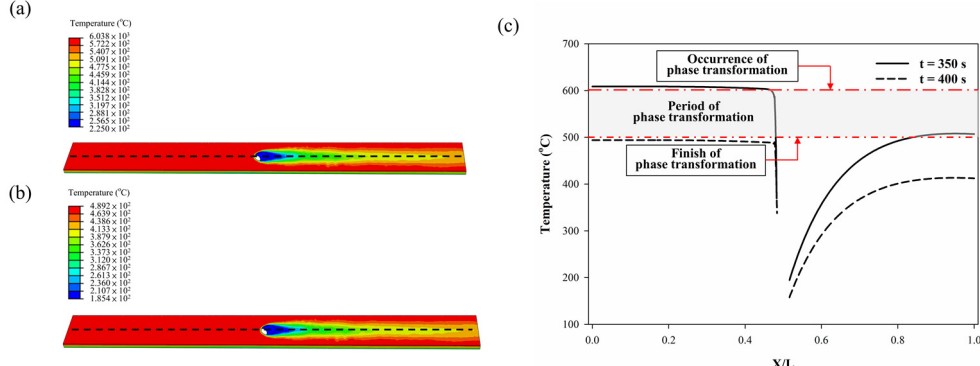

**Figure 6.** Outlined photo for illustrating the overall temperature distribution (**a**) at 350 s, (**b**) 400 s, and (**c**) the temperature distribution along the dashed line at a different time.

Based on Pujol et al. [44], the research on CMAS infiltration, the TC layer under CMAS infiltration was classified into two different layers in the yz plane. One was a CMAS-rich layer, and the other was a non-CMAS layer. Note that the infiltration zone and non-infiltration zone were defined in the xy plane; however, the CMAS-rich layer and non-CMAS layer were outlined in the yz plane. To be more specific, the diagram illustrating the CMAS-rich layer with different penetration depths within the TC layer is shown in Figure 7. In simulation, the penetration depth of the CMAS-rich layer was defined as 35, 70, and 105 μm, as shown in Figure 7a–c, respectively. During the cooling period, the CMAS-rich layer played a critical role owing to the phase transformation. According to previous results [45], the yttria in 7YSZ was depleted by CMAS, which converted the CMAS-rich layer into an yttria-depleted layer. In addition, this CMAS-rich layer was subjected to a phase transformation from the tetragonal phase to the monoclinic phase when the temperature cooled down in the range of transition temperature (500–600 °C). Hence, during this interval of phase transformation (350–400 s), the CMAS-rich layer forcibly implemented a volume expansion of 3%–5% via phase transformation. To verify the influence of volume expansion via phase transformation on the stress field, this study selected a point near the left side of the cooling hole in the infiltration zone to make related observations. The von Mises stress versus time with different expansion rates is shown in Figure 8. The depth of the CMAS-rich layer was assumed to be 70 μm. It was observed that

the von Mises stress smoothly increased owing to contraction during the cooling period when there was no expansion in the CMAS-rich layer during the cooling stage. The stress drastically increased when the expansion rates were 2.5%, 3.5%, and 4.5% from 350 to 400 s. During this phase transformation period (350–400 s), the temperature distribution transformed from 600 to 500 °C as shown in Figure 6c. Specifically, the stress intensified with the increase in the expansion rate. From these results, the volume expansion via phase transformation (tetragonal to monoclinic) was verified to be a critical reason for the premature fracture of TBCs.

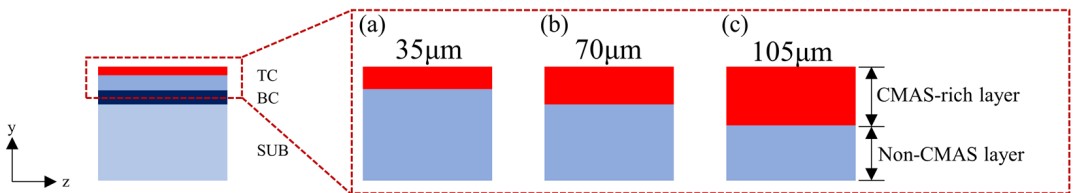

**Figure 7.** Outlined diagram for illustrating (**a**) 35, (**b**) 70, and (**c**) 105 μm penetration depths of CMAS infiltration with the TC layer.

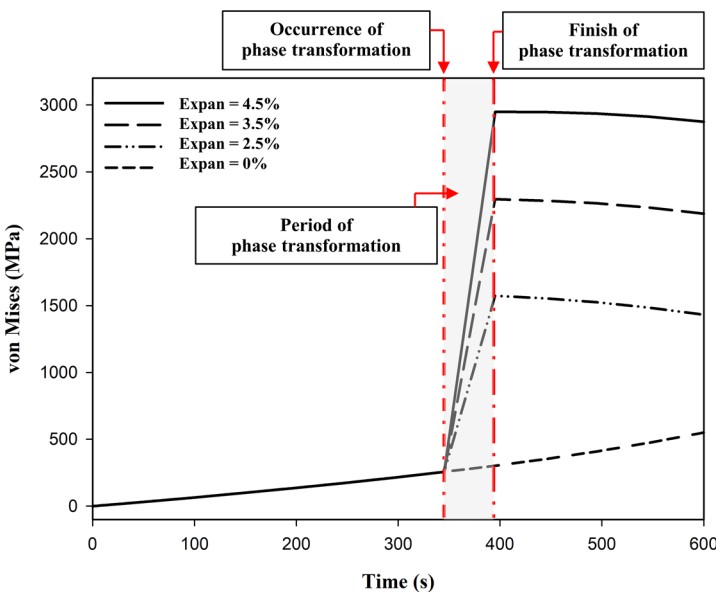

**Figure 8.** Von Mises stress versus time with different expansion rates.

### 3.3. Stress Field with CMAS Infiltration

The gas turbine was affected when volcanic ash accumulated on the surface during the operating period. The volcanic ash melted over a wide range of temperatures and penetrated the TC layer to form a CMAS-rich layer (yttria-depleted layer). Tseng et al. [39,46] and Zhang et al. [47] indicated that the failure life of the TBCs could be influenced by the penetration depth of the CMAS. As mentioned above, the TC layer of the CMAS-rich layer experienced volume expansion because of the phase transformation during cooling period. Thus, the influence of different expansion rates (2.5%, 3.5%, and 4.5%) and penetration depths (0, 35, 70, and 105 μm) have been analyzed under CMAS infiltration. First, the von Mises stress along the BC top surface versus the distance x/L with different expansion rates at room temperature is depicted in Figure 9a. In this case, the depth of the CMAS-rich layer was assumed to be 70 μm, as shown in Figure 7b. The results showed that the von Mises stress gradually decreased when the region was close to the cooling hole and increased when the region was farther away from the hole. This was because the region near the film cooling hole was at a lower temperature during the initial stage and thus experienced smaller residual stress at room temperature. In contrast, the region farther away from

the cooling hole had larger residual stress at room temperature because it was at a higher temperature at the initial stage, as shown in Figure 4a. In conclusion, the stress along the interface between the BC and TC layers was significantly influenced by the expansion rates, as shown in Figure 9a. When the expansion rate via phase transformation during the cooling time was larger, the probability of occurrence of delamination of the TBCs was higher. Specifically, the magnitude of the residual stress became enhanced when the volume expansion was intensified by the phase transformation. From this investigation, volume expansion via phase transformation was an important factor for the failure analysis of TBCs. The von Mises stress versus the distance x/L with different penetration depths along the BC top surface at room temperature is shown in Figure 9b. In this case, the expansion rate was 4.5%. The results showed that the stress was the lowest when the CMAS penetration did not occur, which indicated that the depth of the CMAS-rich layer was zero. However, the von Mises stress was intensified with a larger penetration depth, which implied that the TBCs may have experienced fractures when a large amount of CMAS infiltrated into the TC layer. Further, the stress along the interface between the BC and TC layers was enhanced when CMAS infiltration became deeper. In addition, the stress was concentrated near the cooling hole, as shown in Figure 9a,b.

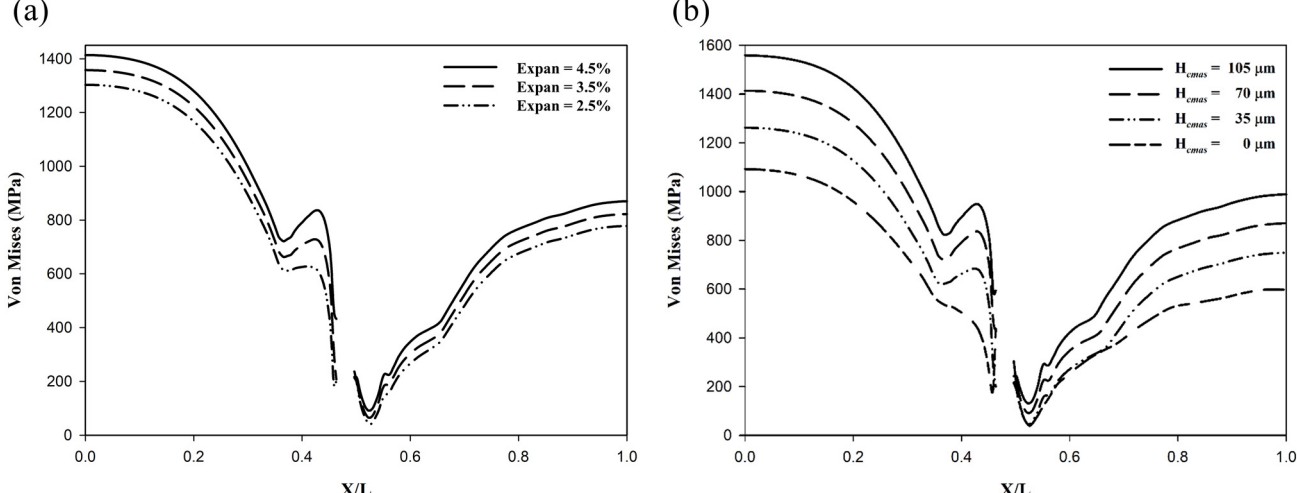

**Figure 9.** Von Mises stress with (**a**) different expansion rates and (**b**) penetration depth along BC top surface.

Interfacial stress is predicted to be the major reason for the delamination of TBCs [11,12]. The boundary conditions around the edge of the film cooling holes in the BC layer, such as the peeling stress, interfacial normal stress ($\sigma_{nn}$), interfacial tangential stress ($\sigma_{tt}$), and interfacial shear stress ($\sigma_{nt}$), have been discussed based on different expansion rates and depth of the CMAS-rich layer. The indicators of different interfacial stresses are shown in Figure 10. The peeling moment and shear force between different layers were derived from the local peeling stress and shear stress, respectively. In addition, the interfacial peeling stress and shear stress resulted in mode-I and mode-II fractures, respectively [18,19]. The interfacial stresses between the BC and TC layers, such as normal stress, tangential stress, and shear stress, were obtained around the cooling hole to predict the fracture mode.

The peeling stress along the BC top surface around the cooling hole with different expansion rates at room temperature is shown in Figure 11a. In this case, the depth of the CMAS-rich layer was assumed to be 70 μm. Note that 0° and 180° were at locations A and B around the cooling hole, respectively, as shown in Figure 5a. As can be seen, the peeling stress displayed an almost zero value at 0°. Location A around the cooling hole (0°) was a non-infiltration zone, and there was no expansion via phase transformation during the cooling period. However, the maximum peeling stress was displayed at 180°, which was at location B around the cooling hole. Location B around the cooling hole

was considered an infiltration zone and was expected to cause mode-I fractures owing to volume expansion due to phase transformation. In addition, location B had the highest temperature at the initial stage, as shown in the upper-left outlined diagram. As mentioned before, the thermal stress was intensified more during the cooling period owing to the large temperature difference. It could be concluded that the largest peeling stress was due to the phase transformation and larger temperature difference.

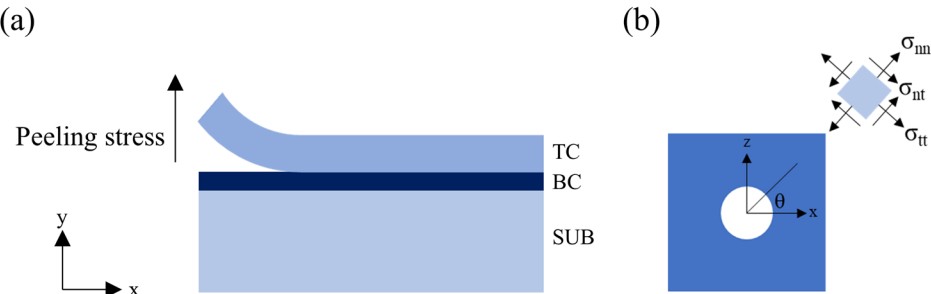

**Figure 10.** Outlined photograph for (**a**) peeling stress and (**b**) interfacial stresses.

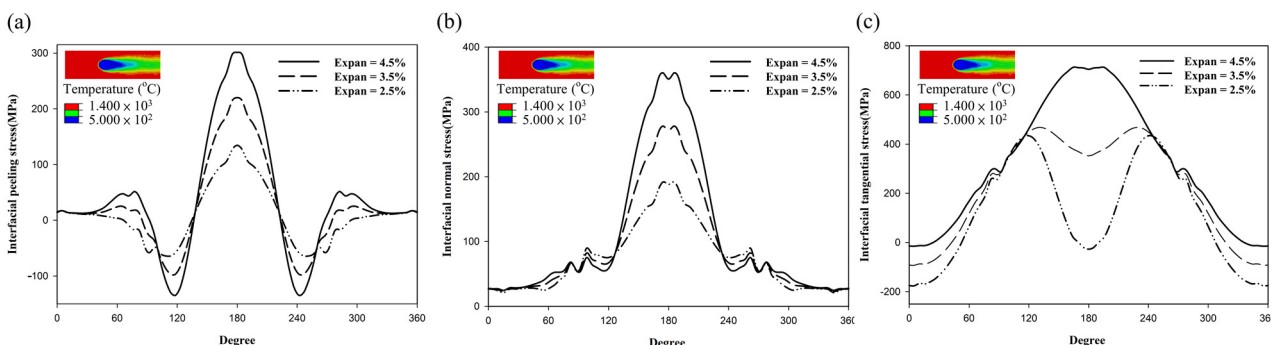

**Figure 11.** (**a**) Peeling stress, (**b**) interfacial normal stress, and (**c**) interfacial tangential stress along the BC top surface around the cooling hole with different expansion rates.

Second, the interfacial normal and tangential stresses near the cooling hole along the BC top surface layer with different expansion rates are shown in Figure 11b,c, respectively. It was found that the distributions of these two stresses appeared to be symmetric. These tendencies of interfacial normal stresses were similar to those of the previous analytically thermoelastic solution for interfacial stresses [48,49]. The maximum normal stress was observed at 180°, and the minimum value was at 0°, as shown in Figure 11b. With regards to the interfacial tangential stress, the maximum tangential stresses with expansion rates of 2.5% and 3.5% were at 120° and 250°, respectively. Furthermore, the maximum tangential stress with a 4.5% expansion rate was found at 180°, as shown in Figure 11c. It was predicted that the failure via interfacial normal and tangential stress was more likely to be noted in location B (infiltration zone) around the cooling hole when the expansion rate was sufficiently large. In addition, when the expansion rate via phase transformation during the cooling time was larger, all the interfacial stresses (peeling, interfacial normal, and tangential) were enhanced. Note that the interfacial shear stress had a relatively small impact on the delamination of the TBCs. Therefore, in the present study, the influence of the interfacial shear stress was ignored.

The influence of different penetration depths on interfacial stress was considered as well. In this case, the expansion rate was assumed to be 4.5%. First, the peeling stress with different-depth CMAS-rich layers along the BC top surface around the cooling hole at room temperature is shown in Figure 12a. The peeling stress drastically increased owing to the infiltration of CMAS compared to that in the TBCs without CMAS infiltration, which implied that TBCs with film cooling holes caused mode-I fractures in location B around

the cooling hole owing to deeper infiltration. The maximum peeling stress was observed at 180°, which implied that the driving force was enhanced at location B, resulting in crack initiation and propagation. Therefore, it was predicted to be the most vulnerable location for mode-I fracture in the infiltration zone near the cooling hole. Second, the interfacial normal and tangential stresses around the cooling hole at the BC layer with different penetration depths are shown in Figure 12b,c, respectively. As shown, the stress distribution was symmetric, and the maximum stress was observed near location B around the cooling hole because of the infiltration zone. It was further predicted that the failure via interfacial normal and tangential stresses was more likely to be observed at location B (infiltration zone) of the cooling hole when CMAS infiltration was sufficiently deep. Based on aforementioned discussions, it could be concluded that fracture of TBCs probably occurred at location B at the infiltration zone owing to the largest interfacial stresses with the larger expansion rate, deeper infiltration, and larger temperature difference. At the beginning of CMAS infiltration (from 0 to 35 μm), the interfacial stress would be more sensitive to the effect of infiltration depth, which indicated that the dramatic increase in the interfacial stresses was found at the beginning of CMAS infiltration.

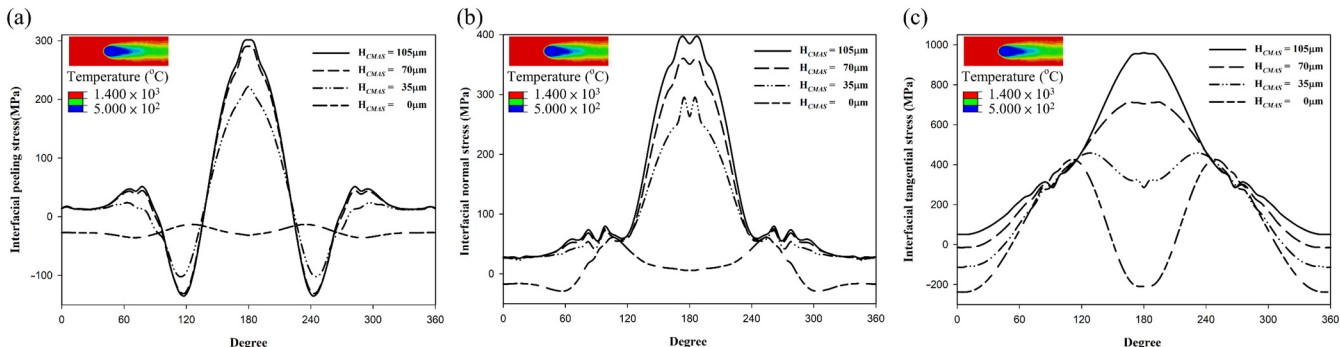

**Figure 12.** (**a**) Peeling stress, (**b**) interfacial normal stress, and (**c**) interfacial tangential stress along the BC top surface around the cooling hole along interfaces with different penetration depths.

## 4. Conclusions

To analyze the premature fracture of TBCs, the effects of film cooling holes and CMAS infiltration were investigated. Thermo-fluid-solid couple analysis was required to define the infiltration zone controlled by the subroutine in ABAQUS. In the end, certain interfacial stresses, such as peeling stress, interfacial normal stress, and interfacial tangential stress, were considered for analyzing the fracture behavior of TBCs. From the results, the influence of the cooling hole was explained by the fact that stress concentration occurred near the cooling hole during cooling, and the presence of a cooling hole suppressed the interfacial von Mises stress by 60% due to the reduction of temperature by 40%. In addition, the effect of CMAS infiltration would enhance residual stresses via phase transformation. Owing to the influence of larger penetration depths and expansion rates via phase transformation, there was a drastic increase in residual stress as a consequence of CMAS infiltration. At the beginning of CMAS infiltration, the interfacial stress would be more sensitive to the effect of infiltration depth. On the other hand, the maximum interfacial stresses appeared at the location near the infiltration zone around the cooling hole. As a result, the infiltration zone near the film cooling hole was predicted to be the fracture site, which resulted in delamination and premature fractures in the TBCs.

**Author Contributions:** Conceptualization, C.C. (Chingkong Chao) and S.T.; methodology, S.T. and X.F.; software, C.C. (Chenchun Chiu), S.T. and W.C.; formal analysis, W.C., S.T. and C.C. (Chenchun Chiu); investigation, C.C. (Chenchun Chiu) and S.T.; resources, C.C. (Chingkong Chao) and X.F.; data curation, C.C. (Chenchun Chiu) and S.T.; writing—original draft preparation, S.T. and C.C. (Chenchun Chiu); writing—review and editing, S.T., C.C. (Chingkong Chao) and X.F.; visualization, S.T., C.C. (Chingkong Chao) and X.F.; project administration, C.C. (Chingkong Chao) and X.F. All authors have read and agreed to the published version of the manuscript.

**Funding:** This study was supported by the National Science and Technology Major Project of China (J2019-IV-0003-0070).

**Institutional Review Board Statement:** Not applicable.

**Informed Consent Statement:** Not applicable.

**Data Availability Statement:** The data presented in this study are available on request from the corresponding author after obtaining permission from the authorized individual.

**Conflicts of Interest:** The authors declare no conflict of interests.

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
