# Peer review of "Interfacial Stresses of Thermal Barrier Coating with Film Cooling Holes Induced by CMAS Infiltration"

_coatings, doi:10.3390/coatings12030326_

Round 1
Reviewer 1 Report
Comment 1: The title should be revised and improved.
Comment 2: Qualitative informations are missing in abstract. Abstract should be concise and the authors need to improve with more specific short results.
Comment 3: The introduction section should be modified though citing recent references (2021 and 2022) related studies and indicating the novelty of the study compared to the carried works.
Comment 4: The figures in the manuscript were poor, the author should improve the quality and solution of these figures.
Comment 5: All equations should be mentioned in the text.
Comment 6: Compare your results with literature ones.
Comment 7: Level of English is good however in a few places some syntax errors are present. At some places two or more words joined together that should be corrected.
Comment 8: Conclusion should be revised and improved.
Author Response
The authors would like to thank all the reviewers who raise important comments and suggestions.

Reviewer 2 Report
Please find the comments in the attached file.

Author Response

(The authors gave the same response as above.)

Reviewer 3 Report
The authors performed three-dimensional numerical simulations to test the thermal performance of the thermal barrier coating system. For that, they use the commercial software ANSYS and ABAQUS. I have the following comments for the authors:
- They must provide the equations they used in their model, and they have to mention why they used such models?
- They must provide dimensional analysis to show which of the parameters involved in the problem are more important than others. Otherwise is very difficult to interoperate their results.
Author Response

(The authors gave the same response as above.)

Round 2
Reviewer 2 Report
authors addressed the comments properly.
Reviewer 3 Report
The only comment I have is that the authors could make their results non-dimensional, so we can understand how the different processes involved affect each other.